# Multi-Label Weighted Contrastive Cross-Modal Hashing

Zeqian Yi, Xinghui Zhu, Runbing Wu, Zhuoyang Zou , Yi Liu and Lei Zhu *

College of Information and Intelligence, Hunan Agricultural University, Changsha 410128, China; yzq@stu.hunau.edu.cn (Z.Y.); zhuxh@hunau.edu.cn (X.Z.); runbingwu@stu.hunau.edu.cn (R.W.); zzy@stu.hunau.edu.cn (Z.Z.); yiliu@hunau.edu.cn (Y.L.)
* Correspondence: leizhu@hunau.edu.cn

**Abstract:** Due to the low storage cost and high computation efficiency of hashing, cross-modal hashing has been attracting widespread attention in recent years. In this paper, we investigate how supervised cross-modal hashing (CMH) benefits from multi-label and contrastive learning (CL) by overcoming the following two challenges: (i) how to combine multi-label and supervised contrastive learning to consider diverse relationships among cross-modal instances, and (ii) how to reduce the sparsity of multi-label representation so as to improve the similarity measurement accuracy. To this end, we propose a novel cross-modal hashing framework, dubbed Multi-Label Weighted Contrastive Hashing (MLWCH). This framework involves compact consistent similarity representation, a new designed multi-label similarity calculation method that efficiently reduces the sparsity of multi-label by reducing redundant zero elements. Furthermore, a novel multi-label weighted contrastive learning strategy is developed to significantly improve hashing learning by assigning similarity weight to positive samples under both linear and non-linear similarities. Extensive experiments and ablation analysis over three benchmark datasets validate the superiority of our MLWCH method, especially over several outstanding baselines.

**Keywords:** cross-modal hashing; contrastive learning; multi-label; compact consistent similarity; similarity weight assigning





## 1. Introduction

With the prosperity of multimedia technology and smart devices, a tremendous number of multi-modal data (e.g., text, image, video, and audio) have been pouring into the Internet [1–5]. Despite differences in structures, various types of data are usually semantically related to each other. These semantic relationships could be used for data retrieval or sharing. Naturally, cross-modal retrieval technology [6–9] has become a desiderata since it efficiently returns one modality as a result to queries of other modalities by effectively mining the intrinsic semantic relationships.

The primary issue of cross-modal retrieval is to reduce the heterogeneity gap between modalities. Most existing approaches try to address this issue by projecting the original features of data into a common real-valued subspace in which the semantic similarity can be easily measured [10–19]. Unfortunately, due to the explosive growth of data, the computational complexity of real-valued cross-modal retrieval has become an unavoidable challenge. A viable solution is cross-modal hashing [20–30], which maps high-dimensional multi-modal features into compact binary codes and the cross-modal similarity can be calculated by XOR operation efficiently.

Depending on whether category information is used during the training stage, existing cross-modal hashing methods are mainly classified into unsupervised and supervised manners. Generally, without category information, it is difficult for unsupervised cross-modal hashing [31–38] to generate cross-modal hash codes with strong semantic discrimination even though they endeavor to learn latent similarity structures among different modalities. Supervised cross-modal hashing [39–45], in contrast, is able to use category labels

to enhance cross-modal semantic discrimination so as to obtain high-quality hash codes. As a special case of supervised hashing learning, multi-label cross-modal hashing methods [29,46–50] aim to efficiently handle the instances associated with multiple labels [51–53]. Unlike traditional single-label hashing methods that focus on binary similarity for individual instances, they use multiple labels to construct a semantic similarity matrix so as to learn more accurate similaritiy relationships (e.g., each similarity is defined as a real value between −1 and 1). Meanwhile, motivated by the remarkable achievements in contrastive learning, some contrastive learning methods [30,54] for cross-modal hashing are introduced which aim to capture cross-modal similarities more effectively by comparing samples across modalities.

**Motivation.** Recently, some great progress has been made in cross-modal hashing with contrastive learning, such as [30,54]. UCCH [54] represents the initial endeavor in employing contrastive learning within unsupervised cross-modal hashing. It introduces a cross-modal ranking learning loss (CRL) to alleviate the influence of false-negative pairs. Conversely, Unihash [30] leverages a contrastive label correlation learning (CLC) loss to establish connections between diverse modalities through category labels. However, they simply apply InfoNCE [55,56] in cross-modal hashing, which treats an image–text pair as a positive sample; otherwise, it is a negative sample. This contrastive learning strategy gives rise to the false-negative problem, where samples belonging to the same class are incorrectly regarded as negative samples, resulting in the learning of erroneous relationships among cross-modal instances. Hence, a widely adopted approach in supervised contrastive learning [57] is to consider instances as similar if they share at least one common category. Taking Figure 1 as an intuitive example, all the semantic similarities between image–text pairs $(v_1, t_1)$, $(v_2, t_2)$, $(v_3, t_3)$ and $(v_4, t_4)$ are considered as 1 due to at least one shared category, i.e., "tree". However, as $v_2$, $v_3$, and $v_4$ share one, three, and four labels with $v_1$, respectively, the semantic similarity between $v_1$ and other instances should be ordered as $S_{12} < S_{13} < S_{14}$, rather than $S_{12} = S_{13} = S_{14} = 1$, where $S$ denotes the semantic similarity matrix. Indisputably, a supervised contrastive learning strategy is not suitable for multi-label scenarios because this naive binary similarity cannot accurately reflect the complex semantic relationships between cross-modal instances. Thus, *the first challenge we have to face is how to combine multi-label and supervised contrastive learning to consider diverse relationships among cross-modal instances.*

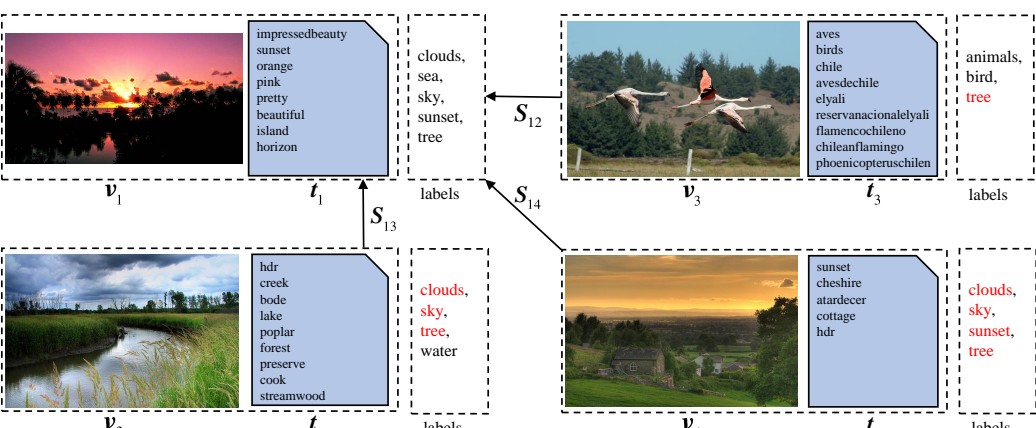

**Figure 1.** The demonstration of semantic similarities between four multi-label image–text instances $\{(v_i, t_i)\}_{i=1}^{4}$, where $S_{12}$, $S_{13}$, $S_{14}$ are the semantic similarities between $(v_1, t_1)$ and $(v_2, t_2)$, $(v_3, t_3)$, and $(v_4, t_4)$, respectively. Depending on the number of labels they share, there should be $S_{12} < S_{13} < S_{14}$. These four image–text pairs are selected from MIRFLICKR-25K. The labels in red represent the shared labels.

More than that, this problem occurs in the similarity matrix construction of most supervised cross-modal methods, i.e., instances are considered similar if they share at least one common category and dissimilar otherwise. Several studies, as pioneers, try to subtly

use shared labels to accurately describe the semantic relationships. For example, ref. [29] uses bi-direction relation reasoning to calculate multi-label similarity in two directions so as to improve semantic similarity matrix construction, where bi-direction relation includes consistent direction relation and inconsistent direction relation. The consistent direction relation is the similarity between two similar instances that share at least one common category, while the inconsistent direction relation refers to the degree of dissimilarity of two instances that do not share any category. However, this method has a glaring flaw; due to the sparsity of multi-labels, as shown in Figure 2a, too many zeros are shared during semantic similarity matrix construction in the consistent direction relation, which makes the method unable to accurately represent the semantic similarity between instances. Thus, *the second challenge we need to overcome is how to efficiently reduce the sparsity of multi-label representation during semantic similarity measurement.*

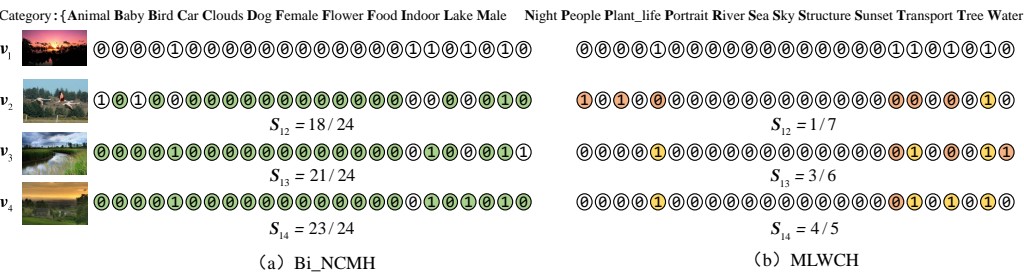

**Figure 2.** Difference between (**a**) Bi_NCMH and (**b**) our method (MLWCH) in multi-label-based consistent direction relation reasoning, where 1 and 0 separately represent the relevant and irrelevant categories. $S_{12}$, $S_{13}$, and $S_{14}$ are the semantic similarities between $v_1$ and $v_2$, $v_3$, and $v_4$ respectively.

**Our Method.** To defeat the above challenges, this paper proposes a novel **M**ulti-**L**abel **W**eighted **C**ontrastive **C**ross-modal **H**ashing (**MLWCH**) method. As shown in Figure 3, on one hand, a novel multi-label similarity measurement, termed *compact consistent similarity representation*, is proposed to improve the accuracy of semantic similarity calculation by producing more compact label vectors. As illustrated in Figure 2b, this technique is capable of reducing the dimensionality of the label vectors by eliminating redundant zero elements, thereby mitigating the potential impact of excessive zeros so as to result in a more compact similarity representation. On the other hand, we extend supervised contrastive learning into multi-label scenarios via a new designed *multi-label weighted contrastive learning* strategy. With the engagement of compact consistent similarity representation, this novel learning strategy assigns different weights to positive samples according to both linear and non-linear similarity relationships.

**Contributions.** The main contributions of this paper are fourfold:

- We develop a novel multi-label cross-modal hashing framework called MLWCH to learn high-quality hash codes. To the best of our knowledge, MLWCH acts as a pioneer in attempting to enhance cross-modal hashing via multi-label contrastive learning supported by more precise semantic similarity representation.
- We propose a novel multi-label similarity measurement, called compact consistent similarity representation, to construct a high-quality semantic similarity matrix. By reducing the sparsity of label vectors through eliminating redundant zero elements, it achieves a more compact similarity calculation and focuses on informative and crucial non-zero elements.
- We design a novel multi-label weighted contrastive learning strategy by marrying supervised contrastive learning with compact consistent similarity representation, which assigns different weights to different positive samples by considering both linear and non-linear similarities.
- We conducted extensive experiments, including performance comparison, ablation study, and hyperparameter sensitivity analysis, on three well-known benchmark datasets. The remarkable results demonstrate the superiority of our method.

**Roadmap.** The rest of the paper is organized as follows. Section 2 reviews the related works. Section 3 presents the details of our multi-label weighted contrastive cross-modal hashing (MLWCH) framework and its optimization. Section 4 shows the evaluation of MLWCH as well as comparison experimental results on three datasets. Section 5 concludes this paper.

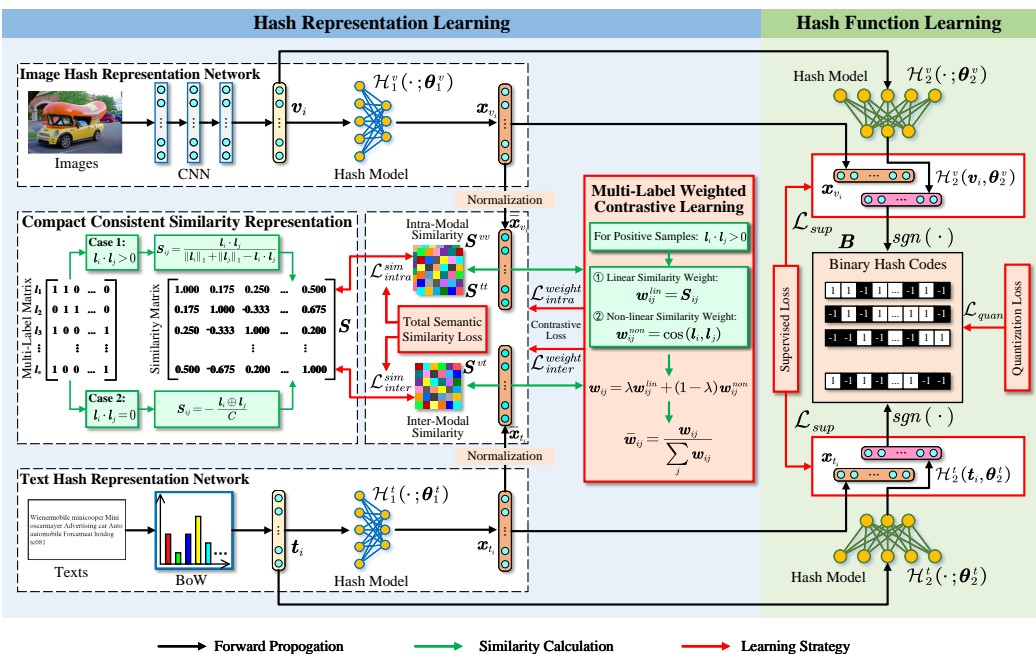

**Figure 3.** The framework of MLWCH. It mainly contains four modules: (i) the compact consistent similarity representation module uses multiple labels to accurately construct the similarity matrix; (ii) the multi-label weighted contrastive learning module realizes a novel contrastive learning strategy in which different positive samples are assigned with different weights based on label sharing; (iii) the hash representation learning module aims to learn high-quality cross-modal hash representations by combining cross-modal semantic similarity learning and weighted contrastive learning; and (iv) the hash function learning module treats the learned hash representation as a supervised signal to preserve semantic consistency. Best view in color.

## 2. Related Work

In this section, we briefly review some related works in terms of supervised cross-modal hashing, unsupervised cross-modal hashing, and contrastive learning.

### 2.1. Supervised Cross-Modal Hashing

By exploiting the discriminative information of labels, almost all supervised cross-modal methods learn modality-specific hash functions to project multi-modal data (such as image–text pairs) into a common Hamming space. For example, DCMH [43] combines hash code learning and feature learning into a unified deep framework. SSAH [44] is a multi-modal semantic learning network that aims to integrate adversarial learning into cross-modal hashing in a self-supervised manner. CMHH [45] utilizes a pairwise focus loss based on an exponential distribution to penalize similar cross-modal pairs whose Hamming distance is greater than a Hamming radius threshold. Bi_NCMH [29] is a bi-directional relational reasoning-based deep cross-modal hashing method, which generates a similarity matrix from multiple labels so as to fully capture the relationship between instances. Due to the semantic information from annotations, the supervised cross-modal hashing methods are able to embed more discriminative semantics into hash codes to achieve promising retrieval performance.

### 2.2. Unsupervised Cross-Modal Hashing

Without any data annotation, unsupervised cross-modal hashing methods learn a unified binary code by minimizing the correlation of cross-modal pairs. Thus, this group of techniques has attracted extensive attention from researchers in both academia and industry. UDCMH [36] combines deep learning and matrix decomposition with binary latent factor models to perform multi-modal data retrieval. DJSRH [58] learns a hash network by utilizing a unified affinity matrix as guidance information, which is generated by combining different forms of semantic similarity. As this combination cannot describe the semantic relationship between instances sufficiently, JDSH [59] is proposed, which is a cross-modal joint training method to maintain cross-modal semantic correlation by constructing a joint-modal similarity matrix. DGCPN [60] utilizes the coherence of neighbors to construct a similarity matrix and guide hashing learning by exploring the relationship between data and neighbors.

### 2.3. Contrastive Learning

Contrastive learning is a type of self-supervised learning method that aims to learn a model to differentiate between similar and dissimilar instances within a dataset. It works by encouraging the model to map similar instances closer together in a learned feature space while pushing dissimilar items farther apart. Recently, contrastive learning [57,61,62] has attracted much attention from the community. For example, ref. [61] observed that apparent similarities can be learned from the data themselves without explicit guidance. Therefore, instead of learning from label-level discrimination, contrastive learning is proposed to learn instance-level discrimination. Supervised contrastive learning [57] proposes a new extension of the contrastive loss function that allows multiple positive samples per anchor, extending contrastive learning to supervised settings. To understand the behavior of contrastive loss, ref. [62] analyzes the relationship between model performance and temperature $\tau$ in contrastive learning, and models with small or large temperatures achieve sub-optimal performance. Inspired by the great success of contrastive learning, some contrastive hashing methods [30,54] have been proposed to learn binary representations from multi-modal data and achieved promising performance. UCCH [54] is the first attempt to use contrastive learning in unsupervised cross-modal hashing, which proposes a cross-modal ranking learning loss (CRL) to mitigate the impact of false-negative pairs. Unihash [30] uses a contrastive label correlation learning (CLC) loss to bridge different modalities by using category labels.

## 3. The Proposed Methodology

In this section, we present our approach, MLWCH, which integrates compact consistent similarity representation and multi-label weighted contrastive learning to generate high-quality cross-modal hash codes. Firstly, we lay out the notational groundwork and give the problem a definition formally, then proceed to our proposed method, including the framework, hash learning strategy, and optimization algorithm.

### 3.1. Notations and Problem Definition

**Notations.** Without loss of generality, sets are denoted as Euler script uppercase letters (e.g., $\mathscr{O}$). Matrices, vectors, and scalars are denoted by bold uppercase letters (e.g., $A$), bold lowercase (e.g., $a$), and uppercase/lowercase (e.g., $N$ or $n$). $A_{ij}$ denotes the $ij$-th element of matrix $A$. The transposition of matrix $A$ is denoted as $A^{\mathrm{T}}$. Vector space with dimensions $n \times m$ is presented as blackboard bold uppercase (e.g., $\mathbb{R}^{n \times m}$). $\|\cdot\|_1$, $\|\cdot\|_2$ and $\|\cdot\|_F$ denotes $L_1$, $L_2$, and Frobenius norm, respectively. Functions or models are denoted as calligraphy uppercase letters (e.g., $\mathcal{H}$). To facilitate reading, the frequently used mathematical notations are summarized in Table 1.

**Table 1.** The summary of frequently used notations.

| Notation | Definition |
|---|---|
| $\mathscr{O}$ | a multi-label cross-modal dataset |
| $N$ | the number of training instances |
| $k$ | length of hash codes |
| $C$ | the number of categories |
| $S$ | semantic similarity matrix of instances |
| $v_i$ | the original feature vector of the $i$th image |
| $t_i$ | the original feature vector of the $i$th text |
| $x_{v_i}$ | the $i$th image hash representation |
| $x_{t_i}$ | the $i$th text hash representation |
| $\bar{x}_{v_i}$ | normalized hash representation of $x_{v_i}$ |
| $\bar{x}_{t_i}$ | normalized hash representation of $x_{t_i}$ |
| $w$ | the similarity weight |
| $\mathcal{H}^v(\cdot, \boldsymbol{\theta}^v)$ | hash model for images |
| $\mathcal{H}^t(\cdot, \boldsymbol{\theta}^t)$ | hash model for texts |
| $B_i$ | the binary hash code of the $i$th instance |

**Problem Definition.** This study considers two commonly used modalities, i.e., image and text. Suppose that there is a multi-label cross-modal dataset $\mathscr{O} = \{o_i\}_{i=1}^{N}$ containing $N$ instances, where $\boldsymbol{o}_i = \{v_i, t_i, l_i\}, i \in \{1, 2, \dots, N\}$ refers to the $i$-th instance; $v_i \in \mathbb{R}^{1 \times d_v}$, $t_i \in \mathbb{R}^{1 \times d_t}$, and $l_i \in \{0, 1\}^C$ are the original image features, text features, and label vectors corresponding to instance $\boldsymbol{o}_i$, respectively. $C$ indicates the number of categories. In particular, if instance $\boldsymbol{o}_i$ is labeled with the $j$-th category, then $l_{ij} = 1$; otherwise, $l_{ij} = 0$. Furthermore, the label vectors can be used to construct an $N \times N$ semantic similarity matrix $S \in \mathbb{R}^{N \times N}$, where $S_{ij} \in [-1, 1]$ indicates the semantic similarity between $l_i$ and $l_j$.

The goal of our work is to learn two hash models $\mathcal{H}^v(\cdot, \boldsymbol{\theta}^v)$ and $\mathcal{H}^t(\cdot, \boldsymbol{\theta}^t)$ (one per modality) on a training dataset $\mathscr{O}$ to generate hash representations from original instances. The element-wise sign function $sgn(\cdot)$ is used to obtain the uniform binary hash code $B \in \{-1, 1\}^{N \times k}$ from continuous hash representations, where $k$ is the code length. In order to conduct cross-modal hashing retrieval, the Hamming distance is involved to measure similarities between instances. Putting it formally, $\forall v_i, t_j, t_k \in \mathscr{O}$, if $S_{ij} \geq S_{ik}$, $Dst_H(B_i, B_j) \leq Dst_H(B_i, B_k)$ and vice versa, where $B_i = sgn(\mathcal{H}^v(v_i, \boldsymbol{\theta}^v))$, $B_j = sgn(\mathcal{H}^t(t_j, \boldsymbol{\theta}^t))$, $B_k = sgn(\mathcal{H}^t(t_k, \boldsymbol{\theta}^t))$. $Dst_H(\cdot, \cdot)$ denotes the Hamming distance between two binary vectors; $\boldsymbol{\theta}^v$ and $\boldsymbol{\theta}^t$ are the model parameters. In this work, the proposed learning framework includes two groups of hash models: one for hash representation learning, denoted as $\mathcal{H}_1^v(\cdot; \boldsymbol{\theta}_1^v)$ and $\mathcal{H}_1^t(\cdot; \boldsymbol{\theta}_1^t)$, and the other for hash function learning, denoted as $\mathcal{H}_2^v(\cdot; \boldsymbol{\theta}_2^v)$ and $\mathcal{H}_2^t(\cdot; \boldsymbol{\theta}_2^t)$.

### 3.2. Overview of MLWCH Framework

As shown in Figure 3, the framework of MLWCH mainly consists of four modules: (i) *the compact consistent similarity representation module*, (ii) *the multi-label weighted contrastive learning module*, (iii) *the hash representation learning module*, and (iv) *the hash function learning module*.

### 3.3. Compact Consistent Similarity Representation

The prevailing solutions of supervised cross-modal hashing typically define a binary semantic similarity relationship. That is, the value of similarity is "0" or "1". This naive manner is unable to accurately measure the complex semantic similarity between two instances. Although it is an advanced similarity calculation, similarity representation in the consistent direction calculated by bi-directional relational reasoning [29] shares too many zeros due to the sparsity of label vectors, which unfortunately hinders accurate similarity measurement between instances. To this end, we propose a novel multi-label semantic similarity measurement called compact consistent similarity representation. As a

refined version of bi-direction relation reasoning, this semantic similarity representation method modifies the similarity calculation in the consistent direction. Specifically, two cases are considered: (i) If two instances $o_i$ and $o_j$ share at least one label, i.e., $l_i \cdot l_j > 0$, their semantic similarity $S_{ij}$ is defined as

$$S_{ij} = \frac{l_i \cdot l_j}{\|l_i\|_1 + \|l_j\|_1 - l_i \cdot l_j},\tag{1}$$

where the $\cdot$ symbol denotes a vector inner product. In Equation (1), the numerator is the number of the shared categories of the $i$-th and $j$-th instances, and the denominator is the number of all categories contained by the $i$-th or $j$-th instance. Therefore, $S_{ij} \in (0,1]$ can measure the similarity between $o_i$ and $o_j$ more accurately than bi-direction relation reasoning since the labels that do not belong to either instance are excluded, which makes similarity measurement intensively focus on semantic relationships caused by shared labels. (ii) If $o_i$ and $o_j$ have no shared label, i.e., $l_i \cdot l_j = 0$, their similarity is defined as

$$S_{ij} = -\frac{l_i \oplus l_j}{C},\tag{2}$$

where $\oplus$ is the XOR operation and $C$ is the number of categories. Thus, $S_{ij} \in [-1,0]$ represents the dissimilarity between $o_i$ and $o_j$.

*3.4. Multi-Label Weighted Contrastive Learning*

The core idea of contrastive learning is maximizing mutual information (MI) between similar instances. MI measures the correlation between two variables and quantifies the amount of information they share. However, in practical applications, the joint distribution and marginal distribution between two variables are often unknown, which is troublesome for MI calculation. Therefore, contrastive learning typically employs some approximate calculation methods. For example, InfoNCE [55,56] provides a low-variance estimate of MI for high-dimensional data. However, InfoNCE is not a supervised learning strategy since it is essentially unable to involve category information. To effectively adapt this learning strategy to multi-label cross-modal hashing, we start with the design of cross-modal supervised contrastive loss, then extend it into a multi-label learning task; as a result, we develop multi-label weighted contrastive loss.

**Cross-Modal Supervised Contrastive Loss.** Inspired by [57], we attempt to extend InfoNCE to a cross-modal supervised learning scenario. Firstly, we select instance pairs that share some categories as positive pairs, and then construct intra-modality InfoNCE loss for both image and text modalities, shown as follows:

$$\mathcal{L}_{intra\text{-}v} = \frac{1}{n}\sum_{i=1}^{n}\frac{1}{n_{l_i}-1}\sum_{j=1}^{n}\mathcal{I}_{i\neq j}\cdot\mathcal{I}_{l_i\cdot l_j>0}\cdot\mathcal{L}_{intra\text{-}v}^{ij},$$

$$\mathcal{L}_{intra\text{-}v}^{ij} = -\log\frac{\exp\left(\bar{x}_{v_i}\cdot\bar{x}_{v_j}/\tau\right)}{\sum_{k=1}^{n}\mathcal{I}_{i\neq k}\cdot\exp\left(\bar{x}_{v_i}\cdot\bar{x}_{v_k}/\tau\right)},\tag{3}$$

$$\mathcal{L}_{intra\text{-}t} = \frac{1}{n}\sum_{i=1}^{n}\frac{1}{n_{l_i}-1}\sum_{j=1}^{n}\mathcal{I}_{i\neq j}\cdot\mathcal{I}_{l_i\cdot l_j>0}\cdot\mathcal{L}_{intra\text{-}t}^{ij},$$

$$\mathcal{L}_{intra\text{-}t}^{ij} = -\log\frac{\exp\left(\bar{x}_{t_i}\cdot\bar{x}_{t_j}/\tau\right)}{\sum_{k=1}^{n}\mathcal{I}_{i\neq k}\cdot\exp\left(\bar{x}_{t_i}\cdot\bar{x}_{t_k}/\tau\right)},\tag{4}$$

where $n_{l_i}$ is the total number of samples that share some common labels with $l_i$, $\mathcal{I}_c$ is a function indicating that if the condition $c$ is true, $\mathcal{I}_c = 1$; otherwise, $\mathcal{I}_c = 0$. Combining the above two loss functions, the intra-modality InfoNCE loss is defined as:

$$\mathcal{L}_{intra} = \mathcal{L}_{intra\text{-}v} + \mathcal{L}_{intra\text{-}t}. \tag{5}$$

In an analogous manner, the inter-modality infoNCE losses for image and text are defined as:

$$\mathcal{L}_{inter\text{-}v} = \frac{1}{n} \sum_{i=1}^{n} \frac{1}{n_{l_i}} \sum_{j=1}^{n} \mathcal{I}_{l_i \cdot l_j > 0} \cdot \mathcal{L}_{inter\text{-}v'}^{ij}$$

$$\mathcal{L}_{inter\text{-}v}^{ij} = -\log \frac{\exp\left(\bar{x}_{v_i} \cdot \bar{x}_{t_j} / \tau\right)}{\sum_{k=1}^{n} \exp\left(\bar{x}_{v_i} \cdot \bar{x}_{t_k} / \tau\right)}, \tag{6}$$

$$\mathcal{L}_{inter\text{-}t} = \frac{1}{n} \sum_{i=1}^{n} \frac{1}{n_{l_i}} \sum_{j=1}^{n} \mathcal{I}_{l_i \cdot l_j > 0} \cdot \mathcal{L}_{inter\text{-}t'}^{ij}$$

$$\mathcal{L}_{inter\text{-}t}^{ij} = -\log \frac{\exp\left(\bar{x}_{t_i} \cdot \bar{x}_{v_j} / \tau\right)}{\sum_{k=1}^{n} \exp\left(\bar{x}_{t_i} \cdot \bar{x}_{v_k} / \tau\right)}. \tag{7}$$

Accordingly, the inter-modality InfoNCE for multi-label cross-modal hashing learning is defined as:

$$\mathcal{L}_{inter} = \mathcal{L}_{inter\text{-}v} + \mathcal{L}_{inter\text{-}t}. \tag{8}$$

Combining Equations (5) and (8), we obtain the cross-modal supervised contrastive loss as follows:

$$\mathcal{L}_{con} = \gamma \mathcal{L}_{intra} + (1 - \gamma) \mathcal{L}_{inter}, \tag{9}$$

where $\gamma \in [0, 1]$ is a trade-off factor of two InfoNCE losses.

**Multi-Label Weighted Contrastive Loss.** Obviously, the above supervised contrastive loss regards all positive samples as equally important. As discussed in Section 1, however, it is not suitable for multi-label scenarios. To break through this limitation, we suppose that different positive samples (i.e., $l_i \cdot l_j > 0$) should be assigned different weights according to shared labels. To this end, a novel multi-label weighted contrastive loss is developed. In particular, two different weights, i.e., linear and non-linear weights, are defined to represent multi-label semantic similarities, shown as follows:

$$w_{ij}^{lin} = S_{ij}, \quad l_i \cdot l_j > 0. \tag{10}$$

$$w_{ij}^{non} = \cos(l_i, l_j). \tag{11}$$

The linear weight $w_{ij}^{lin}$ is to directly capture the relationship between positive samples by counting the shared labels, while the non-linear weight $w_{ij}^{non}$, as a form of complementarity, measures more complex similarity. In addition to the quantity relationship of shared labels, for example, cosine similarity considers the angle and direction relationship between different feature representations as well. Therefore, to comprehensively consider the similarity relationship between instances, we obtain the overall weight $w_{ij}$ via combining Equations (10) and (11) as follows:

$$w_{ij} = \lambda w_{ij}^{lin} + (1 - \lambda) w_{ij}^{non}, \tag{12}$$

where $\lambda \in [0, 1]$ is a trade-off factor to balance the two weights. Then, we normalize the weight as follows:

$$\bar{w}_{ij} = \frac{w_{ij}}{\sum_j w_{ij}}. \tag{13}$$

By introducing the overall weight $\bar{w}_{ij}$, the multi-label weighted contrastive loss functions are defined as follows. For intra-modality:

$$\mathcal{L}_{intra\text{-}v}^{weight} = \frac{1}{n} \sum_{i=1}^{n} \frac{1}{n_{l_i} - 1} \sum_{j=1}^{n} \mathcal{I}_{i \neq j} \cdot \bar{w}_{ij} \cdot \mathcal{L}_{intra\text{-}v}^{ij}, \tag{14}$$

$$\mathcal{L}_{intra\text{-}t}^{weight} = \frac{1}{n} \sum_{i=1}^{n} \frac{1}{n_{l_i} - 1} \sum_{j=1}^{n} \mathcal{I}_{i \neq j} \cdot \bar{w}_{ij} \cdot \mathcal{L}_{intra\text{-}t}^{ij}. \tag{15}$$

For inter-modality:

$$\mathcal{L}_{inter\text{-}v}^{weight} = \frac{1}{n} \sum_{i=1}^{n} \frac{1}{n_{l_i}} \sum_{j=1}^{n} \bar{w}_{ij} \cdot \mathcal{L}_{inter\text{-}v}^{ij}, \tag{16}$$

$$\mathcal{L}_{inter\text{-}t}^{weight} = \frac{1}{n} \sum_{i=1}^{n} \frac{1}{n_{l_i}} \sum_{j=1}^{n} \bar{w}_{ij} \cdot \mathcal{L}_{inter\text{-}t}^{ij}. \tag{17}$$

Compared with traditional supervised contrastive loss, the multi-label weighted contrastive loss assigns a weight to positive samples, which makes the significance of positive samples proportional to their number of shared labels with the anchor sample. Similar to Equations (5) and (8), the intra- and inter-modal weighted contrastive losses are presented as:

$$\mathcal{L}_{intra}^{weight} = \mathcal{L}_{intra\text{-}v}^{weight} + \mathcal{L}_{intra\text{-}t}^{weight}, \tag{18}$$

$$\mathcal{L}_{inter}^{weight} = \mathcal{L}_{inter\text{-}v}^{weight} + \mathcal{L}_{inter\text{-}t}^{weight}. \tag{19}$$

Finally, combining Equations (18) and (19), we obtain the multi-label weighted contrastive loss as follows:

$$\mathcal{L}_{con}^{weight} = \gamma \mathcal{L}_{intra}^{weight} + (1 - \gamma) \mathcal{L}_{inter}^{weight}, \tag{20}$$

where $\gamma \in [0, 1]$ is a trade-off factor to balance the two weighted InfoNCE losses.

*3.5. Hash Representation Learning*

Beyond all doubt, preserving semantic consistency between original instances and their hash representations is a key factor of high-quality hash code generation. In other words, the more similar semantically the instance pair ($o_i$ and $o_j$) is, the smaller the Hamming distance between their hash codes should be, and vice versa. To this end, we construct the hash representation learning objective function by integrating the following three losses: (i) *an intra-modal semantic similarity loss*, (ii) *an inter-modal semantic similarity loss*, and (iii) *a multi-label weighted contrastive loss*.

**Similarity Matrices.** To construct intra- and inter-modal semantic similarity loss, the cross-modal semantic similarities should be represented firstly. Specifically, three similarity matrices, i.e., $S^{vv}, S^{tt}, S^{vt} \in \mathbb{R}^{N \times N}$, are constructed by the inner product between normalized hash representations: $S_{ij}^{vv} = \bar{x}_{v_i}^{\mathrm{T}} \bar{x}_{v_j}$, $S_{ij}^{tt} = \bar{x}_{t_i}^{\mathrm{T}} \bar{x}_{t_j}$, and $S_{ij}^{vt} = \bar{x}_{v_i}^{\mathrm{T}} \bar{x}_{t_j}$, where $S_{ij}^{vv}, S_{ij}^{vt}$ and $S_{ij}^{vt}$ are the semantic similarity between $v_i$ and $v_j$, and $t_i$ and $t_j$, as well as $v_i$ and $t_j$, respectively.

**Intra-Modal Semantic Similarity Loss.** According to the above similarity matrices, we define the intra-modal semantic similarity loss for both modalities to preserve intra-modal semantic consistency:

$$\mathcal{L}_{intra\text{-}v}^{sim} = \sum_{i,j=1}^{N} \left\| S_{ij} - S_{ij}^{vv} \right\|_F^2, \tag{21}$$

$$\mathcal{L}_{intra\text{-}t}^{sim} = \sum_{i,j=1}^{N} \left\| S_{ij} - S_{ij}^{tt} \right\|_F^2. \tag{22}$$

**Inter-Modal Semantic Similarity Loss.** To preserve inter-modal semantic consistency, we use the inter-modal semantic similarity loss to effectively capture the heterogeneous similarities across different modalities:

$$\mathcal{L}_{inter}^{sim} = \sum_{i,j=1}^{N} \left\| S_{ij} - S_{ij}^{vt} \right\|_F^2. \tag{23}$$

Combining Equations (21) and (22), we obtain the intra-modal semantic similarity loss as follows:

$$\mathcal{L}_{intra}^{sim} = \mathcal{L}_{intra\text{-}v}^{sim} + \mathcal{L}_{intra\text{-}t}^{sim}. \tag{24}$$

Combining Equations (23) and (24), we obtain the total semantic similarity loss as follows:

$$\mathcal{L}_{total}^{sim} = \mathcal{L}_{inter}^{sim} + \mathcal{L}_{intra}^{sim}. \tag{25}$$

Finally, we construct the hash representation learning objective function $\mathcal{L}_{rep}$ that consists of the above two losses, shown as follows:

$$\mathcal{L}_{rep} = \mathcal{L}_{con}^{weight} + \alpha \mathcal{L}_{total}^{sim}, \tag{26}$$

where $\alpha \in [0,1]$ is a trade-off factor.

### 3.6. Hash Function Learning

We learn another two modality-specific hash models, i.e., $\mathcal{H}_2^v(\cdot, \boldsymbol{\theta}_2^v)$ and $\mathcal{H}_2^t(\cdot, \boldsymbol{\theta}_2^t)$, to generate a binary hash code from each instance $\boldsymbol{o}_i \in \mathscr{O}$ in the following manner:

$$B_i = sgn\left( \frac{\mathcal{H}_2^v(\boldsymbol{v}_i, \boldsymbol{\theta}_2^v) + \mathcal{H}_2^t(\boldsymbol{t}_i, \boldsymbol{\theta}_2^t)}{2} \right), \tag{27}$$

where $sgn(\cdot)$ is the sign function; $B_i$ denotes the binary hash codes of instance $\boldsymbol{o}_i$. Then, we treat the learned hash representation as a supervised signal to guide the hash function learning:

$$\mathcal{L}_{sup} = \sum_{i=1}^{N} \left\| \mathcal{H}_2^v(\boldsymbol{v}_i, \boldsymbol{\theta}_2^v) - \boldsymbol{x}_{v_i} \right\|_2^2 + \left\| \mathcal{H}_2^t(\boldsymbol{t}_i, \boldsymbol{\theta}_2^t) - \boldsymbol{x}_{t_i} \right\|_2^2. \tag{28}$$

To reduce quantization error, the following quantization loss is involved:

$$\mathcal{L}_{quan} = \sum_{i=1}^{N} \left\| B_i - \mathcal{H}_2^v(\boldsymbol{v}_i, \boldsymbol{\theta}_2^v) \right\|_2^2 + \left\| B_i - \mathcal{H}_2^t(\boldsymbol{t}_i, \boldsymbol{\theta}_2^t) \right\|_2^2. \tag{29}$$

Combining Equations (28) and (29), we obtain the total loss for hash function learning as follows:

$$\mathcal{L}_{func} = \mathcal{L}_{sup} + \mathcal{L}_{quan}. \tag{30}$$

### 3.7. Optimization

The learning process of the proposed MLWCH consists of two stages: (i) the hash representation learning stage and (ii) the hashing function learning stage. We adopt the Adam adaptive algorithm [63] for optimization. In the hash representation learning stage, we minimize Equation (26) to optimize the hash representation learning model $\mathcal{H}_1^v(\cdot, \boldsymbol{\theta}_1^v)$ and $\mathcal{H}_1^t(\cdot, \boldsymbol{\theta}_1^t)$ as follows:

$$(\hat{\boldsymbol{\theta}}_1^v, \hat{\boldsymbol{\theta}}_1^t) = \arg\min_{\boldsymbol{\theta}_1^v, \boldsymbol{\theta}_1^t} \mathcal{L}_{rep}. \tag{31}$$

In the hash function learning stage, we minimize Equation (30) to optimize the hash function learning model $\mathcal{H}_2^v(\cdot, \boldsymbol{\theta}_2^v)$ and $\mathcal{H}_2^t(\cdot, \boldsymbol{\theta}_2^t)$ as follows:

$$(\hat{\boldsymbol{\theta}}_2^v, \hat{\boldsymbol{\theta}}_2^t) = \underset{\boldsymbol{\theta}_2^v, \boldsymbol{\theta}_2^t}{arg\min} \mathcal{L}_{func}. \tag{32}$$

Overall, the entire optimization procedure of MLWCH is presented in Algorithm 1.

---

**Algorithm 1** Optimization procedure for MLWCH.

---

**Input:** Number of training image–text pairs $N$; number of epochs $e$; hash code length $k$;
    batch size $b$; learning rate of the network $\eta_1$ and $\eta_2$; hyperparameter $\alpha, \lambda, \gamma, \tau$;
**Output:** Optimized parameters $\hat{\boldsymbol{\theta}}_1^v, \hat{\boldsymbol{\theta}}_1^t, \hat{\boldsymbol{\theta}}_2^v, \hat{\boldsymbol{\theta}}_2^t$.

 1: Construct a semantic similarity matrix $S$ from multi-label set $\{l_i\}_{i=1}^N$;
 2: **repeat**
 3:    Iterative training $e$ times;
 4:    // Hash representation learning stage: optimizing objective function Equation (31)
 5:    **for** each $i \in \left[1, \left\lceil \frac{N}{b} \right\rceil\right]$ **do**
 6:        Randomly select $b$ training image–text pairs;
 7:        Generate continuous hash representation $x_v, x_t$ through $\mathcal{H}_1^v(\cdot, \boldsymbol{\theta}_1^v), \mathcal{H}_1^v(\cdot, \boldsymbol{\theta}_1^t)$;
 8:        Calculate the loss $\mathcal{L}_{rep}$ by Equation (26) and update the parameters $\boldsymbol{\theta}_1^v, \boldsymbol{\theta}_1^t$ through
           back propagation as follows:

$$\boldsymbol{\theta}_1^v \longleftarrow \boldsymbol{\theta}_1^v - \eta_1 \times \frac{\partial}{\partial \boldsymbol{\theta}_1^v}\left(\mathcal{L}_{con}^{weight} + \alpha \mathcal{L}_{total}^{sim}\right)$$
$$\boldsymbol{\theta}_1^t \longleftarrow \boldsymbol{\theta}_1^t - \eta_1 \times \frac{\partial}{\partial \boldsymbol{\theta}_1^t}\left(\mathcal{L}_{con}^{weight} + \alpha \mathcal{L}_{total}^{sim}\right)$$

 9:    **end for**
10:    // Hash function learning stage: optimizing objective function Equation (32)
11:    **for** each $i \in \left[1, \left\lceil \frac{N}{b} \right\rceil\right]$ **do**
12:        select $b$ training image–text pairs and hash representation $x_v, x_t$;
13:        Map the original feature $v$ and $t$ into a hash code through $\mathcal{H}_2^v(\cdot, \boldsymbol{\theta}_2^v), \mathcal{H}_2^t(\cdot, \boldsymbol{\theta}_2^t)$;
14:        Calculate the loss $\mathcal{L}_{func}$ by Equation (30), and update the parameter $\boldsymbol{\theta}_2^v, \boldsymbol{\theta}_2^t$ through
           back propagation as follows:

$$\boldsymbol{\theta}_2^v \longleftarrow \boldsymbol{\theta}_2^v - \eta_2 \times \frac{\partial}{\partial \boldsymbol{\theta}_2^v}\left(\mathcal{L}_{sup} + \mathcal{L}_{quan}\right)$$
$$\boldsymbol{\theta}_2^t \longleftarrow \boldsymbol{\theta}_2^t - \eta_2 \times \frac{\partial}{\partial \boldsymbol{\theta}_2^t}\left(\mathcal{L}_{sup} + \mathcal{L}_{quan}\right)$$

15:    **end for**
16: **until** epoch $== e$

---

## 4. Experiment

To evaluate the performance of the proposed method holistically, extensive experiments are carried out on three widely used cross-modal retrieval datasets: MIRFLICKR-25K [64], NUS-WIDE [65], and MS COCO [66]. This section firstly introduces the experimental settings, including datasets, evaluation metrics, baselines, and implementation details. Then, we delve into the performance comparison of MLWCH with the baselines, the ablation study, and the hyperparameter sensitivity analysis.

### 4.1. Datasets

**MIRFLICKR-25K.** The original MIRFLICKR-25K is made up of 25,000 image–text pairs from the Flickr website. In our experiment, we remove those pairs that have fewer than 20 tags, and finally obtain 20,015 image–tag pairs. Then, we extract 4096-dimensional CNN (AlexNet [67]) features to represent each image, and 1386-dimensional Bag-of-Words (BoW) [68] features to represent each text.

**NUS-WIDE.** The original NUS-WIDE dataset contains 269,468 image–text pairs. We first abandon the data without categories, then choose data classified by the 10 most frequent categories to construct a subset which has 186,577 image–text pairs. For our

experiments, we encode each image into a 4096-dimensional feature by AlexNet and each textual tag into a 1000-dimensional BoW feature.

**MS COCO.** This dataset contains 123,287 image–text pairs in 80 independent categories in total. Similar to the above datasets, a 4096-dimensional feature vector is generated by AlexNet for each image, and a BoW model is adopted to represent its corresponding text with 2000 dimensions.

### 4.2. Evaluation Metrics

To objectively evaluate the performance of our proposed method and compare it with the baseline methods, two frequently used cross-modal hashing evaluation protocols, i.e., Hamming ranking and hash lookup [69], are utilized in our experiments. The former ranks samples in the retrieval set by their Hamming distance to the query in ascending order, while the latter retrieves samples within a certain Hamming radius from the query [43]. The mean average precision (MAP) is used to measure the accuracy of the Hamming ranking protocol, while precision-recall curves (PR curves) are commonly used to measure the accuracy of the hash lookup protocol. Given a query $q_k$, the AP score of top $n$ results is calculated by

$$AP(q_k) = \sum_{i=1}^{n} \frac{\mathcal{I}(i)}{N} \sum_{j=1}^{i} \frac{\mathcal{I}(j)}{i}, \tag{33}$$

where $\mathcal{I}(i)$ is an indicator function; if the $i$th retrieved sample is similar to the query, i.e., sharing at least one common category with the query, $\mathcal{I}(i) = 1$; otherwise, $\mathcal{I}(i) = 0$. $N$ denotes the number of relevant samples in the returned top $n$ samples. MAP is the average of APs for all queries:

$$MAP = \frac{1}{K} \sum_{k=1}^{K} AP(q_k), \tag{34}$$

where $K$ is the size of the query set.

Moreover, during evaluation, an image $v_i$ and a text $t_j$ will be treated as a similar pair if they share at least one common label.

### 4.3. Baselines and Implementation Details

**Baselines.** We compare the proposed MLWCH method with nine classical or state-of-the-art cross-modal hashing methods, including three shallow model-based methods, (CVH [40], SCM [41], and CCA-ITQ [70]), and six deep model-based methods, (DCMH [43], SSAH [44], DCHUC [26], SCCGDH [27]), MMACH [28], and Bi_NCMH [29]. A brief introduction of each is presented below:

- CVH aims to learn a hash function to efficiently find similar data items in the hash space by mapping data from different views to hash codes.
- SCM introduces a large-scale supervised multi-modal hashing approach that emphasizes the idea of semantic relevance maximization for efficient similarity search between different modal data.
- CCA-ITQ introduces an iterative quantization method to gradually improve the quality of binary codes through multiple iterations.
- DCMH is the first attempt to integrate deep learning and hash learning into a unified framework for end-to-end learning.
- SSAH is a self-supervised adversarial hashing method which treats labels as a single modality to supervise the learning process of semantic features and integrates adversarial learning into cross-modal hashing in a self-supervised manner.
- DCHUC utilizes an iterative learning optimization algorithm to jointly learn hash codes and hash functions, where the learned hash codes and functions can supervise each other during the optimization process.

- SCCGDH is a class-specific center-guided deep hashing method which makes use of hash codes of labels generated from labeled networks for class-specific centers and efficiently guides hash learning for image and text modalities.
- MMACH integrates a new multi-label modality augmented attention module with self-supervised learning to supervise the training of hash functions for image and text modalities based on augmented multi-labels.
- Bi_NCMH is a bi-directional relational reasoning-based deep cross-modal hashing method that builds a multi-label semantic similarity matrix through consistent and inconsistent relationships between instances.

For a fair comparison, following [27], we utilize AlexNet pre-trained on ImageNet for extracting image features, and employ the Bag-of-Words (BoW) model for extracting text features. For MMACH and Bi_NCMH, as their source code was not available, we carefully implemented these methods.

**Implementation Details.** As shown in Figure 3, each of our hash models is composed of a two-layer multi-layer perceptron with $tanh(\cdot)$, formally represented as: $\mathcal{H}^v(\cdot, \boldsymbol{\theta}^v)$ is $(d_v \rightarrow 512 \rightarrow k)$ and $\mathcal{H}^t(\cdot, \boldsymbol{\theta}^t)$ is $(d_t \rightarrow 512 \rightarrow k)$.

Three hyperparameters, $\lambda$, $\gamma$, and $\tau$, are involved in multi-label weighted contrastive learning. As introduced above, $\lambda$ is the trade-off factor between linear and non-linear weight, $\gamma$ plays the trade-off between intra- and inter-modality weighted contrastive loss, and $\tau$ is the temperature coefficient in contrastive learning. For hash representation learning, we use $\alpha$ to adjust the importance of $\mathcal{L}_{con}^{weight}$ and $\mathcal{L}_{total}^{sim}$. In our experiments, for the parameter $\lambda$, we assign values of 0.3, 0.6, and 0.2 for MIRFlickr-25K, NUS-WIDE, and MS COCO, respectively. Regarding the parameter $\gamma$, we set it to 0.1 for all datasets. Similarly, for the parameter $\tau$, we set it to 0.4, 0.46, and 0.26 for MIRFlickr-25K, NUS-WIDE, and MS COCO, respectively. Furthermore, for the parameter $\alpha$, we assign values of 0.4, 0.2, and 0.1 for MIRFlickr-25K, NUS-WIDE, and MS COCO, respectively. The Adam optimization algorithm [63] is adopted for model training, and we set the learning rate of the hash representation learning $\eta_1$ and hash function learning $\eta_2$ to 0.001 and 0.0001, respectively, on MIR Flickr-25K; 0.001 and 0.0001, respectively, on NUS-WIDE; and 0.0015 and 0.0005, respectively, on MS COCO. The batch size is set to 512. For all experiments, two cross-modal retrieval tasks are considered: *I2T* and *T2I*, where *I2T* represents the cases when using a querying image while returning text, and *T2I* represents the cases when using a querying text while returning an image.

**Experimental Environment.** All experiments were implemented using Python 3.8 on PyTorch 1.12.1 framework, running on a deep learning workstation with Intel(R) Core i9-12900K 3.9 GHz, 128 GB RAM, 1 TB SSD and 2 TB HDD storage, and 2 NVIDIA GeForce RTX 3090Ti GPUs with Ubuntu-22.04.1 operating system.

*4.4. Performance Comparisons and Discussion*

We investigate the retrieval performance of the proposed method MLWCH by comparing it with several state-of-the-art baselines on the MIRFLICKR-25K, NUS-WIDE, and MS COCO datasets. In the following, we discuss the comparison via Hamming ranking and hash lookup.

**Hamming Ranking.** The MAP@50 of MLWCH and baseline methods under distinct hash code lengths 16 bits, 32 bits, and 64 bits are listed in Tables 2–4. From the experimental results, we have the following findings:

- The proposed method MLWCH has shown remarkable performance on all benchmark datasets: it beats both the hand-crafted methods and the deep neural network-based methods in all cases. Particularly, our method outperforms SCCGDH, the strongest competitors, by a significant margin. For the I2T task, the results were 0.0339 (16 bits), 0.0392 (32 bits), and 0.0524 (64 bits) on MIRFLICKR-25K; 0.0335 (16 bits), 0.0174 (32 bits), and 0.0364 (64 bits) on NUS-WIDE; and 0.0755 (16 bits), 0.0499 (32 bits), and 0.0495 (64 bits) on MS COCO. For the T2I task, the results were 0.0708 (16 bits), 0.0425 (32 bits), and 0.0386 (64 bits) on MIRFLICKR-25K; 0.0217 (16 bits), 0.0231 (32 bits), and

0.0268 (64 bits) on NUS-WIDE; and 0.0609 (16 bits), 0.0801 (32 bits), and 0.0609 (64 bits) on MS COCO. These outstanding results verify that integrating compact consistent similarity representation with multi-label weighted contrastive learning can effectively enhance the performance of cross-modal hashing retrieval.

- It is clear to see that the deep hashing methods achieve superior retrieval performances than traditional shallow hashing methods in most cases on the three datasets. The main reason may be that deep learning methods can extract more essential high-level features than traditional shallow methods, which effectively reduces the semantic gap between modalities.

- It can be found that other than the proposed method, all these baselines also achieve relatively lower results on MS COCO than the other two datasets. This observation is mainly due to the fact that the MS COCO dataset provides more label categories than the two other datasets, which in other words brings greater challenges to cross-modal hash learning. Nevertheless, MLWCH still achieves the best results, which corroborates that the proposed technique could efficiently capture cross-modal semantic consistency in complex semantic conditions.

- Compared to deep cross-modal hashing methods utilizing multi-labels, our proposed method MLWCH still obtains the highest performance. Particularly, it in all cases greatly defeats Bi_NCMH, which uses bi-direction relation reasoning. The superiority of MLWCH is partly due to the fact that the proposed technique realizes a more accurate multi-label similarity consistency reasoning to calculate the semantic relevance of original instances. In addition, the combination of multi-label learning with supervised contrastive learning can effectively minimize the heterogeneity gap of original instances, which delivers superior hash learning performance.

**Hash Lookup.** By varying the Hamming radius from 0 to $k$, we plot the PR curves for hash code lengths of 16 bits, 32 bits, and 64 bits on the MIRFLICKR-25K, NUS-WIDE, and MS COCO datasets, respectively. These curves are depicted in Figures 4–6. It is obvious that, in all cases, the PR curves of the proposed method are evidently higher than those of all baselines, which verifies that MLWCH can learn cross-modal semantic relationships more efficaciously than the prevailing solutions.

**Table 2.** The MAP@50 results of our method and baselines varied with different hash code lengths on MIRFLICKR-25K. The best results are bold-font.

| Methods | MIRFLICKR-25K | | | | | |
| | I2T | | | T2I | | |
| | 16 Bits | 32 Bits | 64 Bits | 16 Bits | 32 Bits | 64 Bits |
| --- | --- | --- | --- | --- | --- | --- |
| CVH | 0.5981 | 0.5988 | 0.6190 | 0.6151 | 0.6263 | 0.6311 |
| CCA-ITQ | 0.7814 | 0.7939 | 0.8043 | 0.7509 | 0.7594 | 0.7708 |
| SCM | 0.6800 | 0.6903 | 0.6972 | 0.6853 | 0.6909 | 0.7433 |
| DCMH | 0.7911 | 0.8128 | 0.8288 | 0.8092 | 0.8355 | 0.8293 |
| SSAH | 0.8409 | 0.8607 | 0.8781 | 0.8081 | 0.8314 | 0.8413 |
| DCHUC | 0.8190 | 0.8236 | 0.8299 | 0.8177 | 0.8296 | 0.8417 |
| Bi_NCMH | 0.8521 | 0.8704 | 0.8895 | 0.8192 | 0.8464 | 0.8541 |
| MMACH | 0.8652 | 0.8808 | 0.8923 | 0.8112 | 0.8395 | 0.8672 |
| SCCGDH | 0.8817 | 0.8918 | 0.8926 | 0.8204 | 0.8562 | 0.8716 |
| MLWCH (Ours) | **0.9216** | **0.9310** | **0.9450** | **0.8912** | **0.8987** | **0.9102** |

**Table 3.** The MAP@50 results of our method and baselines varied with different hash code lengths on NUS-WIDE. The best results are bold-font.

| Methods | NUS-WIDE | | | | | |
| --- | --- | --- | --- | --- | --- | --- |
| | I2T | | | T2I | | |
| | 16 Bits | 32 Bits | 64 Bits | 16 Bits | 32 Bits | 64 Bits |
| CVH | 0.6202 | 0.6360 | 0.6358 | 0.6174 | 0.6265 | 0.6352 |
| CCA-ITQ | 0.5955 | 0.6823 | 0.7303 | 0.6523 | 0.6584 | 0.6745 |
| SCM | 0.6587 | 0.6913 | 0.7072 | 0.6556 | 0.6898 | 0.7015 |
| DCMH | 0.7076 | 0.7248 | 0.7272 | 0.6835 | 0.6982 | 0.7130 |
| SSAH | 0.7103 | 0.7240 | 0.7794 | 0.6894 | 0.6858 | 0.6869 |
| DCHUC | 0.7432 | 0.7626 | 0.7678 | 0.6816 | 0.6726 | 0.6968 |
| Bi_NCMH | 0.7417 | 0.7593 | 0.7963 | 0.7120 | 0.7212 | 0.7303 |
| MMACH | 0.7426 | 0.7645 | 0.8054 | 0.7074 | 0.7135 | 0.7274 |
| SCCGDH | 0.8124 | 0.8372 | 0.8401 | 0.7669 | 0.7734 | 0.7722 |
| MLWCH (Ours) | **0.8459** | **0.8546** | **0.8765** | **0.7886** | **0.7965** | **0.7990** |

**Table 4.** The MAP@50 results of our method and baselines varied with different hash code lengths on MS COCO. The best results are bold-font.

| Methods | MS COCO | | | | | |
| --- | --- | --- | --- | --- | --- | --- |
| | I2T | | | T2I | | |
| | 16 Bits | 32 Bits | 64 Bits | 16 Bits | 32 Bits | 64 Bits |
| CVH | 0.3611 | 0.3497 | 0.3587 | 0.4267 | 0.4340 | 0.4262 |
| CCA-ITQ | 0.3504 | 0.3931 | 0.4120 | 0.4006 | 0.4124 | 0.4106 |
| SCM | 0.3697 | 0.3803 | 0.4161 | 0.4296 | 0.4383 | 0.4614 |
| DCMH | 0.5549 | 0.6053 | 0.6085 | 0.5723 | 0.5916 | 0.5986 |
| SSAH | 0.6203 | 0.6231 | 0.6390 | 0.5839 | 0.6203 | 0.6323 |
| DCHUC | 0.6138 | 0.6454 | 0.6595 | 0.5608 | 0.5903 | 0.6284 |
| Bi_NCMH | 0.6857 | 0.7174 | 0.7442 | 0.6454 | 0.6836 | 0.7229 |
| MMACH | 0.7013 | 0.7345 | 0.7564 | 0.6761 | 0.7058 | 0.7323 |
| SCCGDH | 0.7317 | 0.7772 | 0.8084 | 0.7741 | 0.7813 | 0.8315 |
| MLWCH (Ours) | **0.8072** | **0.8271** | **0.8579** | **0.8350** | **0.8614** | **0.8924** |

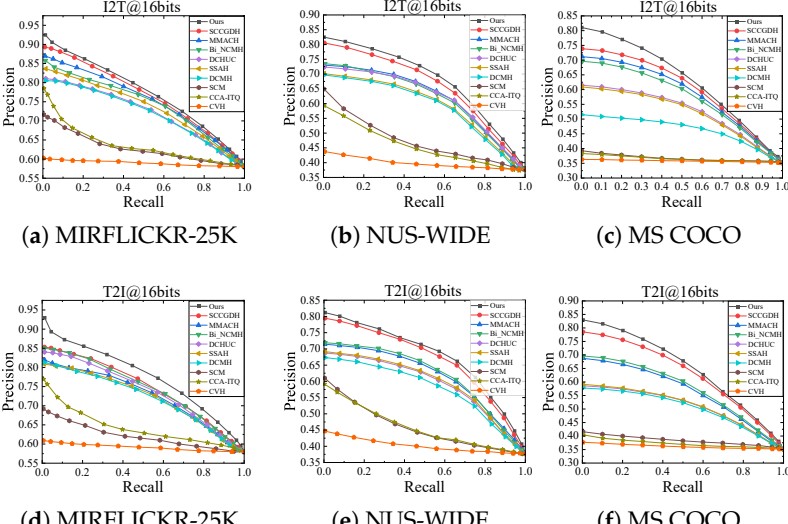

(**a**) MIRFLICKR-25K    (**b**) NUS-WIDE    (**c**) MS COCO

(**d**) MIRFLICKR-25K    (**e**) NUS-WIDE    (**f**) MS COCO

**Figure 4.** Precision-recall curves on MIRFLICKR-25K, NUS-WIDE, and MS COCO datasets. The code length is 16 bits.

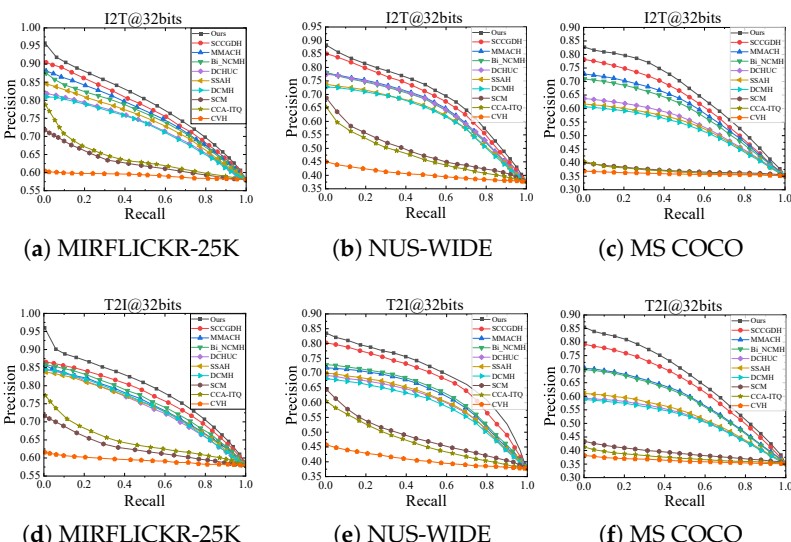

(**a**) MIRFLICKR-25K  (**b**) NUS-WIDE  (**c**) MS COCO

(**d**) MIRFLICKR-25K  (**e**) NUS-WIDE  (**f**) MS COCO

**Figure 5.** Precision-recall curves on MIRFLICKR-25K, NUS-WIDE, and MS COCO datasets. The code length is 32 bits.

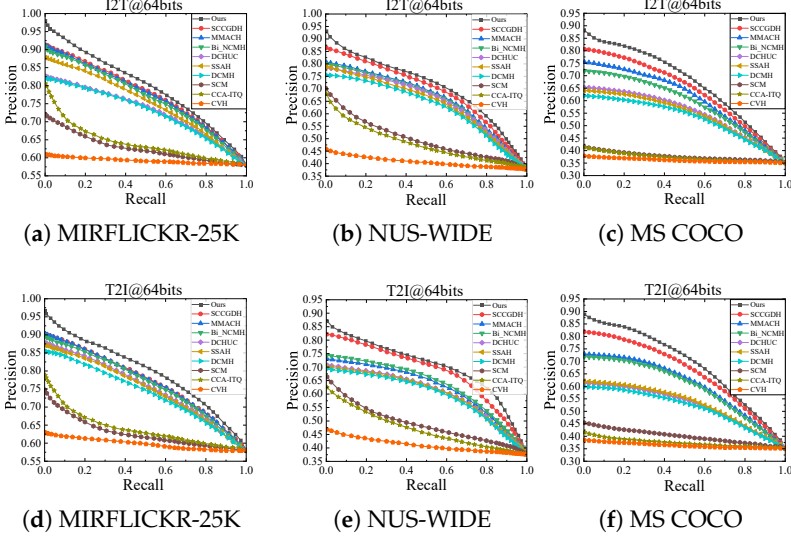

(**a**) MIRFLICKR-25K  (**b**) NUS-WIDE  (**c**) MS COCO

(**d**) MIRFLICKR-25K  (**e**) NUS-WIDE  (**f**) MS COCO

**Figure 6.** Precision-recall curves on MIRFLICKR-25K, NUS-WIDE, and MS COCO datasets. The code length is 64 bits.

### 4.5. Ablation Study

As mentioned above, compact consistent similarity representation and multi-label weighted contrastive learning are two instrumental components of MLWCH. To fully evaluate the effectiveness of MLWCH, we conducted a thorough ablation study to analyze the effectiveness of these two techniques, respectively.

#### 4.5.1. Effectiveness of Compact Consistent Similarity Representation

We evaluate the effectiveness of the compact consistent similarity representation module. Before experiment, we vary MLWCH method by the following two ways. Firstly, we replace the multi-label similarity measurement with a naive manner that is commonly used by prevailing solutions: If the cross-modal samples share at least one label, the similarity is 1; otherwise, it is 0. For the sake of discussion, this variation is named SLWCH. Secondly, we replace the multi-label similarity matrix in MLWCH with the multi-label similarity matrix in Bi_NCMH [29] and keep other parts unchanged. This variation is termed MLBRH. Afterward, we compare the hashing performance of MLWCH with SLWCH and MLBRCH

on MIRFLICKR-25K, NUS-WIDE, and MS COCO. The corresponding MAP@50 values for hash code lengths of 16, 32, and 64 on the three datasets are presented in Tables 5, 6, and 7, respectively.

**Table 5.** Performance of MLWCH compared with SLWCH and MLBRCH in terms of MAP@50 on MIRFLICKR-25K. The best results are bold-font.

| Methods | MIRFLICKR-25K | | | | | |
| | I2T | | | T2I | | |
| | 16 Bits | 32 Bits | 64 Bits | 16 Bits | 32 Bits | 64 Bits |
|---|---|---|---|---|---|---|
| SLWCH | 0.8993 | 0.9074 | 0.9160 | 0.8681 | 0.8721 | 0.8763 |
| MLBRH | 0.9042 | 0.9183 | 0.9331 | 0.8817 | 0.8860 | 0.8904 |
| MLWCH | **0.9216** | **0.9310** | **0.9450** | **0.8912** | **0.8987** | **0.9102** |

**Table 6.** Performance of MLWCH compared with SLWCH and MLBRCH in terms of MAP@50 on NUS-WIDE. The best results are bold-font.

| Methods | NUS-WIDE | | | | | |
| | I2T | | | T2I | | |
| | 16 Bits | 32 Bits | 64 Bits | 16 Bits | 32 Bits | 64 Bits |
|---|---|---|---|---|---|---|
| SLWCH | 0.8303 | 0.8452 | 0.8655 | 0.7764 | 0.7852 | 0.7879 |
| MLBRH | 0.8384 | 0.8508 | 0.8717 | 0.7808 | 0.7898 | 0.7924 |
| MLWCH | **0.8459** | **0.8546** | **0.8765** | **0.7886** | **0.7965** | **0.7990** |

**Table 7.** Performance of MLWCH compared with SLWCH and MLBRCH in terms of MAP@50 on MS COCO. The best results are bold-font.

| Methods | MS COCO | | | | | |
| | I2T | | | T2I | | |
| | 16 Bits | 32 Bits | 64 Bits | 16 Bits | 32 Bits | 64 Bits |
|---|---|---|---|---|---|---|
| SLWCH | 0.7984 | 0.8184 | 0.8496 | 0.8262 | 0.8521 | 0.8834 |
| MLBRH | 0.8036 | 0.8233 | 0.8538 | 0.8307 | 0.8579 | 0.8880 |
| MLWCH | **0.8072** | **0.8271** | **0.8579** | **0.8350** | **0.8614** | **0.8924** |

As can be seen from Tables 5–7, on one hand, the MAP@50 of MLBRH and MLWCH are higher than that of SLWCH in all cases, which indicates that compared with single-label semantic similarity, the complex semantic relationships between instances are more accurately captured by multi-label semantic similarity. On the other hand, the performance of MLWCH is overall superior to MLBRH on three datasets, which confirms that the proposed compact consistent similarity representation module efficaciously improves the accuracy of cross-modal hashing retrieval. This is mainly because the proposed technique is instrumental in modeling sparse multi-labels so as to capture the semantic similarities between instances more accurately, which is a characteristic not possessed by Bi_NCMH [29], in contrast.

### 4.5.2. Effectiveness of Multi-Label Weighted Contrastive Learning

To verify the effectiveness of the proposed multi-label weighted contrastive learning, we make a variation of MLWCH, named MLSCH, by replacing multi-label weighted contrastive loss in Equation (20) with the loss in Equation (9). To compare MLSCH with MLWCH, we also consider the MAP@50 metrics of them under the different hash code lengths. The results on MIRFLICKR-25K, NUS-WIDE, and MS COCO are given in Tables 8, 9, and 10, respectively.

For both I2T and T2I tasks, as manifested in Tables 8–10, the retrieval accuracy of MLWCH is higher than that of MLSCH, which confirms that, guided by the proposed

loss $\mathcal{L}_{con}^{weight}$, the quality of cross-modal hash code learning can be improved noticeably. The main reason behind this phenomenon is that this novel contrastive learning strategy enables the model to perceive more precise similarity relationships by assigning different weights to different positive instances.

**Table 8.** Performance of MLWCH compared with MLSCH in terms of MAP@50 on MIRFLICKR-25K. The best results are bold-font.

| Methods | MIRFLICKR-25K | | | | | |
| | I2T | | | T2I | | |
| | 16 Bits | 32 Bits | 64 Bits | 16 Bits | 32 Bits | 64 Bits |
|---|---|---|---|---|---|---|
| MLSCH | 0.9079 | 0.9234 | 0.9377 | 0.8756 | 0.8916 | 0.8992 |
| MLWCH | **0.9216** | **0.9310** | **0.9450** | **0.8912** | **0.8987** | **0.9102** |

**Table 9.** Performance of MLWCH compared with MLSCH in terms of MAP@50 on NUS-WIDE. The best results are bold-font.

| Methods | NUS-WIDE | | | | | |
| | I2T | | | T2I | | |
| | 16 Bits | 32 Bits | 64 Bits | 16 Bits | 32 Bits | 64 Bits |
|---|---|---|---|---|---|---|
| MLSCH | 0.8365 | 0.8457 | 0.8634 | 0.7803 | 0.7894 | 0.7934 |
| MLWCH | **0.8459** | **0.8546** | **0.8765** | **0.7886** | **0.7965** | **0.7990** |

**Table 10.** Performance of MLWCH compared with MLSCH in terms of MAP@50 on MS COCO. The best results are bold-font.

| Methods | MS COCO | | | | | |
| | I2T | | | T2I | | |
| | 16 Bits | 32 Bits | 64 Bits | 16 Bits | 32 Bits | 64 Bits |
|---|---|---|---|---|---|---|
| MLSCH | 0.7990 | 0.8212 | 0.8509 | 0.8298 | 0.8567 | 0.8861 |
| MLWCH | **0.8072** | **0.8271** | **0.8579** | **0.8350** | **0.8614** | **0.8924** |

### 4.6. Hyperparameter Sensitivity Analysis

This section sheds light on the sensitivity of $\lambda$, $\gamma$, $\tau$, and $\alpha$ on the MIRFLICKR, NUS-WIDE, and MS COCO datasets. To explore the comprehensive impact of them on hashing learning, the average accuracy of I2T and T2I tasks is used to visualize the trend of cross-modal hashing performance. All these analyses are conducted with hash code length 16.

#### 4.6.1. Trade-Off Parameter $\lambda$

As shown in Equation (12), the hyperparameter $\lambda$ is a trade-off factor of linear and non-linear weights. We observe the performance change of MLWCH by varying $\lambda$. From Figure 7, when we set $\lambda$ to 0.3, 0.6, and 0.2 on MIRFLICKR-25K, NUS-WIDE, and MS COCO, respectively, our method obtains the best performance. Under this circumstance, the cross-modal hashing model focuses almost equally on linear and non-linear semantic similarity between instances. Undoubtedly, this manner is essential for comprehensively expressing complex cross-modal semantic relationships.

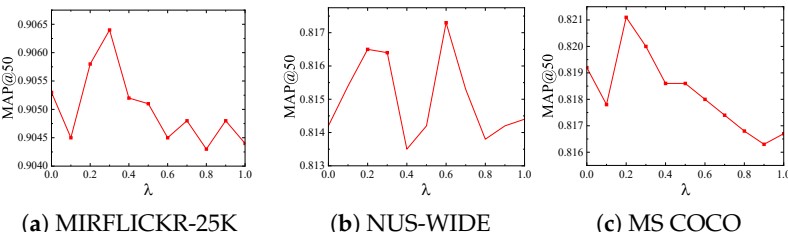

**Figure 7.** Sensitivity analysis of $\lambda$ on MIRFLICKR-25K, NUS-WIDE and MS COCO.

### 4.6.2. Trade-Off Parameter $\gamma$

As presented in Equation (20), hyperparameter $\gamma$ is to balance the two components of multi-label weighted contrastive loss, i.e., intra- and inter-modal weighted InfoNCE losses $\mathcal{L}_{intra}^{weight}$ and $\mathcal{L}_{inter}^{weight}$. To analyze the effect by $\mathcal{L}_{intra}^{weight}$ and $\mathcal{L}_{inter}^{weight}$, we recorded the performance change by varying the value of $\gamma$ on MIRFLICKR-25K, NUS-WIDE and MS COCO datasets. Figure 8 reports that our method MLWCH achieves the best MAP score when $\gamma$ is set to 0.1. Albeit $\mathcal{L}_{intra}^{weight}$ can constrain the intra-modality semantic structure, the performance gain is relatively minor for the cross-modal hashing task. In contrast, inter-modal weighted InfoNEC loss has a relatively greater effect on hashing learning. We conjecture that this result is mainly because inter-modal contrastive strategy is more important for eliminating cross-modal heterogeneity. The above results also clarify that by selecting an appropriate value of $\gamma$, our model can achieve impressive performance.

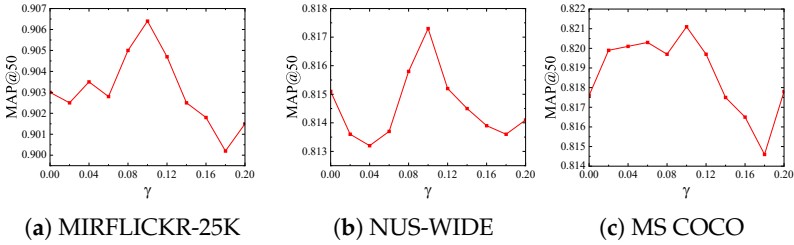

**Figure 8.** Sensitivity analysis of $\gamma$ on MIRFLICKR-25K, NUS-WIDE, and MS COCO.

### 4.6.3. Temperature Parameter $\tau$

As mentioned in previous work [62], the temperature hyperparameter $\tau$ can obviously affect the model performance in contrastive learning. To test and verify this viewpoint on the proposed multi-label weighted contrastive learning, we analyze the hyperparameter $\tau$ of sensitivity on the MIRFLICKR-25K, NUS-WIDE, and MS COCO datasets. Figure 9 illustrates the effect of $\tau$ in MLWCH on these datasets, respectively, making it easy to see that the comprehensive retrieval accuracy varies significantly with the change of $\tau$, and MLWCH achieves the best MAP scores when $\tau$ is set to 0.4, 0.46, and 0.26 on MIRFLICKR-25K, NUS-WIDE, and MS COCO, respectively. This supports the claim that by selecting an appropriate value of $\tau$, our method can achieve the best performance.

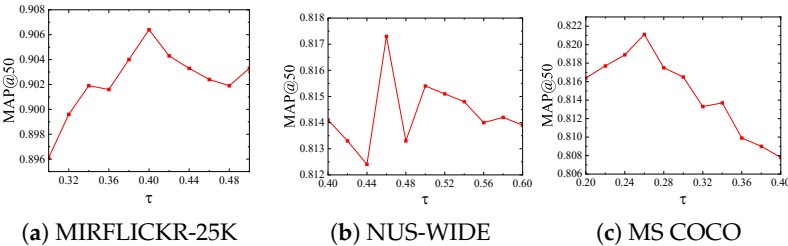

**Figure 9.** Sensitivity analysis of $\tau$ on MIRFLICKR-25K, NUS-WIDE, and MS COCO.

### 4.6.4. Trade-Off Parameter $\alpha$

As depicted in Equation (26), the hyperparameter $\alpha$ serves as a trade-off factor between the multi-label weighted contrastive loss and the semantic similarity loss. From the observations in Figure 10, it is evident that by setting $\alpha$ to 0.4, 0.2, and 0.1 on the MIRFLICKR-25K, NUS-WIDE, and MS COCO datasets, respectively, our method attains the optimal performance. This result demonstrates the effectiveness of incorporating both the semantic similarity loss and the multi-label weighted contrastive learning loss. By carefully selecting an appropriate value for $\alpha$, the proposed method can achieve superior performance.

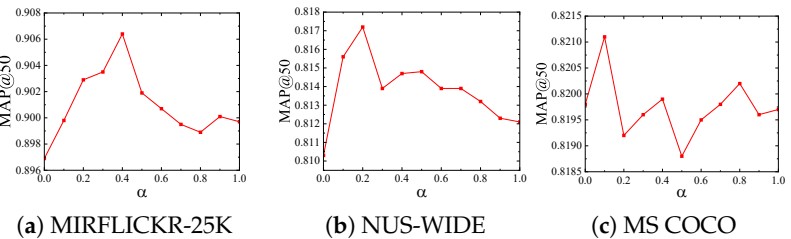

| (a) MIRFLICKR-25K | (b) NUS-WIDE | (c) MS COCO |

**Figure 10.** Sensitivity analysis of $\alpha$ on MIRFLICKR-25K, NUS-WIDE, and MS COCO.

## 5. Conclusions and Future Work

In this paper, we proposed an effective approach called the multi-label weighted contrastive cross-modal hashing (MLWCH) method for high-quality cross-modal hash code generation. A novel compact consistent similarity representation module is designed to accurately preserve the multi-label semantic similarity of original instances. Furthermore, a multi-label weighted contrastive learning strategy is introduced to minimize the gap of the semantic similarity of original instances and more accurately preserve the semantic relevance of the learned hash representations. Extensive experiments on three well-known cross-modal datasets show that the proposed method significantly outperforms other baselines and achieves state-of-the-art performance.

Moving forward, we will endeavor to extend the contrast learning strategy of assigning different weights to different positive samples to unsupervised cross-modal hashing. As an exploratory work, we will investigate the allocation of distinct weights to positive samples through unsupervised learning in scenarios where label information is unavailable.

**Author Contributions:** Conceptualization, L.Z.; methodology, Z.Y. and L.Z.; software, Z.Y.; validation, Z.Y., R.W. and Z.Z.; formal analysis, L.Z.; investigation, X.Z. and Z.Y.; resources, X.Z. and L.Z.; data curation, R.W. and Z.Z.; writing—original draft preparation, Z.Y.; writing—review and editing, L.Z. and Y.L.; visualization, R.W. and Z.Z.; supervision, X.Z. and L.Z.; project administration, X.Z.; funding acquisition, X.Z. and L.Z. All authors have read and agreed to the published version of the manuscript.

**Funding:** This work was supported in part by the National Natural Science Foundation of China (62202163, 62072166), the Natural Science Foundation of Hunan Province (2022JJ40190), the Scientific Research Project of Hunan Provincial Department of Education (22A0145), and the Key Research and Development Program of Hunan Province (2020NK2033).

**Institutional Review Board Statement:** Not applicable.

**Informed Consent Statement:** Not applicable.

**Data Availability Statement:** The public datasets used in this paper can be accessed through the following links: MIRFLICKR-25K: https://press.liacs.nl/mirflickr/mirdownload.html, accessed on 19 December 2023; NUS-WIDE: https://lms.comp.nus.edu.sg/wp-content/uploads/2019/research/nuswide/NUS-WIDE.html, accessed on 19 December 2023; MS COCO: https://cocodataset.org, accessed on 19 December 2023.

**Conflicts of Interest:** The authors declare no conflict of interest.

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
