# Peer review of "Multi-Label Weighted Contrastive Cross-Modal Hashing"

_applsci, doi:10.3390/app14010093_

Round 1

Reviewer 1 Report

Comments and Suggestions for Authors

This paper addresses the challenges of cross-model hashing in a multi-label scenario. The authors employ a contrastive learning strategy, as well as a weighting strategy to learn linear and nonlinear relationships across the modalities. I found the proposed approach interesting, and their conclusion seems supported by the result presented. I do have some concerns with the practicality of the approach because it has a lot of hyperparameters. However, the authors do provide an ablation study to show the effect of each hyperparameter value.   I found the paper to be a little lengthy, I would encourage the authors to revise the paper to have a shorter length without compromising the contents presented. There are also missing references and citations, which need to be addressed.

1. What is the main question addressed by the research?

This paper addresses the challenges of cross-model hashing in a multi-label scenario. The authors employ a contrastive learning strategy, as well as a weighting strategy to learn linear and nonlinear relationships across the modalities.

2. Do you consider the topic original or relevant in the field?

Does it address a specific gap in the field? I think the work is original, and think the authors are addressing the specific gap of handling multi-label cross modal hashing.

3. What does it add to the subject area compared with other published
material?

They introduce a novel cross modal hashing algorithm with contrastive learning.

4. What specific improvements should the authors consider regarding the
methodology? What further controls should be considered?

I think their methodology is fine, but their writeup is hard to follow, especially the mathematical notations. If they sampled the notations that would greatly increase the readability of the paper.
Overall paper reads a little verbose. e.g. line 82-100, 203-219, the presentation of the paper can be improved by quite a bit if the authors refine their wordings. I think at least 2-3 pages can be removed from the paper this way and the presentation can be significantly improved.
There are missing citations and references that need to be fixed for the publication to get accepted.

5. Are the conclusions consistent with the evidence and arguments presented and do they address the main question posed?

Yes.

6. Are the references appropriate?

I think so.

7. Please include any additional comments on the tables and figures.

The figures are small and hard to follow, I would suggest the authors increase their size (e.g. Figure 4 onward). Figure 1 and 2 titles could be more descriptive to describe to better describe their idea.

Comments on the Quality of English Language

I think the writeup is OK and I found the article well organized.

Reviewer 2 Report

Comments and Suggestions for Authors

The authors have investigated how supervised cross-modal hashing (CMH) benefits from multi-label and contrastive learning (CL) by overcoming the following two challenges: (i) how to combine supervised multi-label and contrastive learning to take into account the various relationships between cross-modal instances, and (ii) how to reduce the sparsity of the multi-label representation to improve the accuracy of the similarity measure. The authors' work appears to be very consistent and well-presented. 

I thoroughly enjoyed reading it and found a certain maturity in the presentation of the work carried out. However, I do have a few comments that could help to improve it further: 

1- The title of Figure 1 seems a little ambiguous and difficult: s12, s13 and S13 have the following similarities: (v1, t1)-(v2, t2), (v1, t1)-(v3, t3), and (v1, t1)-(v4, t4)?

2- Figure 2: replace ours with the name of your method. Same as remark 1.

3- In the motivation paragraph: "... with contrastive learning, such as [30,54]" before quoting papers 30 and 54 please name the relevant progress.

4- Section 2.3: please begin by clearly defining the concept of "contrastive learning".

5- Why use distance as a similarity metric (cosine, Manhattan, etc.) and not another type? 

6- Section 4.2: there's a quotation error, and why is line 277 so long? The same error is in section 4.3 (line 284) and the rest of the paper.

7- Analyze the complexity (and if possible explicability) of your models and introduce this parameter into the comparisons and discussions that are made.

Comments on the Quality of English Language

No comment.

Reviewer 3 Report

Comments and Suggestions for Authors

This paper, “Multi-label weighted contrastive cross-modal hashing”.

In my opinion this paper addresses a topic of great importance nowadays. The results obtained seem to be very promising, the solution to the problem has adequate mathematical support. However, I have some comments:

1.      In line 205, I have found a small mistake; “conpact” instead of Compact.

2.      “As discussed in section??” be careful.

3.      “Hamming ranking and hash lookup [?]” be careful.

4.      In line 284 [?] be careful.

5.      Verify the whole paper to avoid these types of mistakes.

I recommend performing an ANOVA analysis on the results obtained from comparing the proposed model with the other methods, since in some the differences are very small. This can mean that maybe it is not the best. An ANOVA analysis gives better arguments for a good conclusion.
